# Mass-Spectrometry-Based Functional Proteomic and Phosphoproteomic Technologies and Their Application for Analyzing Ex Vivo and In Vitro Models of Hypertrophic Cardiomyopathy

**DOI:** 10.3390/ijms222413644

**Published:** 2021-12-20

**Authors:** Jarrod Moore, Andrew Emili

**Affiliations:** 1Center for Network Systems Biology, Boston University School of Medicine, Boston, MA 02118, USA; jmoore5@bu.edu; 2Department of Biochemistry, Boston University School of Medicine, Boston, MA 02118, USA; 3MD-PhD Program, Boston University School of Medicine, Boston, MA 02118, USA

**Keywords:** functional proteomics, mass spectrometry, cardiac disease modeling, hypertrophic cardiomyopathy

## Abstract

Hypertrophic cardiomyopathy (HCM) is an autosomal dominant disease thought to be principally caused by mutations in sarcomeric proteins. Despite extensive genetic analysis, there are no comprehensive molecular frameworks for how single mutations in contractile proteins result in the diverse assortment of cellular, phenotypic, and pathobiological cascades seen in HCM. Molecular profiling and system biology approaches are powerful tools for elucidating, quantifying, and interpreting dynamic signaling pathways and differential macromolecule expression profiles for a wide range of sample types, including cardiomyopathy. Cutting-edge approaches combine high-performance analytical instrumentation (e.g., mass spectrometry) with computational methods (e.g., bioinformatics) to study the comparative activity of biochemical pathways based on relative abundances of functionally linked proteins of interest. Cardiac research is poised to benefit enormously from the application of this toolkit to cardiac tissue models, which recapitulate key aspects of pathogenesis. In this review, we evaluate state-of-the-art mass-spectrometry-based proteomic and phosphoproteomic technologies and their application to in vitro and ex vivo models of HCM for global mapping of macromolecular alterations driving disease progression, emphasizing their potential for defining the components of basic biological systems, the fundamental mechanistic basis of HCM pathogenesis, and treating the ensuing varied clinical outcomes seen among affected patient cohorts.

## 1. Introduction

Hypertrophic cardiomyopathy (HCM) is the most common inherited cardiomyopathy, affecting up to 1 in 500 people [1]. HCM is an autosomal dominant, monogenic disease, thought to be principally caused by mutations in sarcomeric proteins. Roughly 60% of patients have a defined genetic disease, with the majority of mutations lying in thick and thin myofilaments proteins [2,3]. These mutations include singly substituted residues (such as in myosin-7, MYH7), as well as truncated proteins (myosin-binding protein C, MYBPC3) [2]. Despite well-defined mutations, a high degree of variation is observed in clinical phenotypes. MYH7 and MYBPC3 patients often develop left ventricular hypertrophy, fibrosis, and microvascular occlusion, though patients harboring other mutations are observed to have low to no cardiac remodeling [2]. While this may be explained by environmental differences and pleiotropic manifestations of the same mutation, low correlation between mutation and clinical prognosis suggests these mutations are not the sole source of clinical phenotypes [4]. It is plausible that a combination of molecular cascades secondary to these genetic perturbations, as well as compensatory mechanisms, drive the cellular and clinical phenotypes observed. The challenge of mapping such cascades is amplified when considering the complexity of molecular functions performed by cardiomyocytes (CM), the stroma, and other associated cardiac cells.

Molecular profiling and system biology approaches are powerful tools for elucidating and quantifying dynamic signaling pathways and differential expression profiles for a wide range of sample types. There are now high-throughput omics technologies for quantifying nearly all macromolecules, including RNA and proteins, as well those for capturing physical interactions among biomolecules (e.g., protein–protein and protein–DNA interactions). The application of these tools in cardiac research increasingly combines high-performance analytical instrumentation (e.g., mass spectrometry) with computational methods (e.g., bioinformatics) to measure relative molecule abundance and infer the comparative activity of molecular pathways based on coordinated changes in functionally linked molecules [5]. Mass-spectrometry-based methods can also measure post-translational modifications, such as phosphorylation of proteins, which are important signaling processes in cardiovascular disease. [6]. These data can be subject to statistical enrichment analyses and be used to determine the differential activity of phospho-signaling transduction pathways for a given condition or treatment.

Concurrently, cardiac research is benefiting enormously from advancements in in vitro and ex vivo cardiac tissue modeling that recapitulate key aspects of pathogenesis (Figure 1A). Researchers now have an assortment of sources for cardiac cells, each with its own benefits and challenges for modeling cardiac disease progression. For example, differentiation of human-induced pluripotent stem cells into CMs (hiPSC-CMs) provides an abundant source of cells for functional studies, but these cells typically display an immature fetal-like cellular phenotype. Conversely, primary CMs (both human and murine) better reflect adult CM function but are more challenging to obtain in large numbers for experimentation. Undoubtedly, coupling cardiac tissue modeling techniques with quantitative functional proteomic profiling technology offers the potential to generate high-throughput platforms for dissecting, and potentially reversing, the complex signaling cascades altered in HCM.

In this review, we evaluate state-of-the-art mass-spectrometry-based proteomic and phosphoproteomic technology and their application for the analysis of in vitro and ex vivo HCM models, emphasizing their potential role in understanding the fundamental basis of HCM pathogenesis and the ensuing varied clinical outcomes.

## 2. Mass-Spectrometry-Based Proteomics and Phosphoproteomic Technologies

### 2.1. Fundamentals of Mass Spectrometry

LC–MS/MS offers a flexible set of strategies for detecting, quantifying, and identifying diverse macromolecules (e.g., proteins) based on differences in their biophysical properties. In the most basic iteration, a typical LC–MS/MS experiment encompasses four components: sample isolation (e.g., biochemical purification/fractionation), ionization, mass measurement, and bioinformatic analysis (Figure 1B,C). An analogy for this process is using a prism to resolve different wavelengths of light. White light is composed of numerous photon wavelengths (mixed spectrum), and by applying a filter (the prism), each packet of wavelengths can be deconvoluted and individually observed as a band of a specific color [7]. During mass spectrometry (MS) analysis, an individual signal of sample components (e.g., distinct peptides) is deconvoluted from others in complex mixtures (e.g., cell lysate) with a mass filter, such as an electromagnetic field to separate ionized components based on differences in their mass to charge ratio (*m/z*).

Sample preparation prior to the LC–MS/MS analysis is dependent on the type of molecule class measured [8]. Proteins are usually processed via bottom-up proteomics, in which they are proteolytically digested with a protease prior to the LC–MS/MS analysis (Figure 1B). Among other possible methods, they are extracted by lysing cells with a denaturant, shearing DNA and other macromolecules by sonication, and isolating peptides by reversed-phase resins. Once isolated, complex sample mixtures are subject to chromatographic fractionation. This fractionation further deconvolutes mixtures based on differences in component column retention time (e.g., size or biochemical properties), simplifying the complexity of the specimen and allowing the mass spectrometer to view fewer overlapping molecules at any one time. These simplified components are then injected into the MS instrument, in which mass analysis is completed using either a data-dependent or a data-independent data acquisition procedure (DDA and DIA, respectively) (Figure 1B).

With DDA analysis, ions entering the instrument are accumulated within a mass analyzer and quickly scanned (profiled collectively), and then a subset is selected for analysis based on their relative intensity (e.g., top 10 most intense peptide peaks). The instrument is usually programmed to automatically select the highest intensity ions (or precursors) inside a pre-selected *m/z* range, measuring their mass precisely in the first round of a tandem MS analysis. The machine then isolates and energetically fragments individual molecules in a second round of analysis (Figure 1B) [9].

In contrast, DIA can provide more comprehensive analyses since all ions that accumulate in the mass analyzer within a pre-selected *m/z* range are selected jointly for fragmentation [10]. While the accurate masses of the precursor ions and their fragmentation patterns are used for the identification of the original precursor, a challenge arises from the sheer complexity of the mixed MS2 spectra, since many ions of diverse origin are observed at once. DIA spectral deconvolution requires special and increasingly sophisticated computational methods, which are still being optimized [11].

### 2.2. Proteomics and Phosphoproteomics

Proteins are ubiquitous actors of normal and pathological cardiac physiology. Cardiac excitation–contraction is mediated by channels and myofilament proteins that facilitate depolarization and filament sliding. Moreover, cellular cascades such as adrenergic signaling cause rapid changes in the phospho-signaling process [12]. Given the myriad of cellular processes that occur in a cardiac cell, identifying these regulatory relationships and their respective role in the aberrant phenotype in cardiomyopathies is a significant challenge. Historically, such molecules were identified and analyzed via targeted biochemical assays, such as Western blot analysis, which can be hard to scale up for discovery purposes [13]. In contrast, MS-based proteomics and phosphoproteomics (herein, both will be referred to together as proteomics) serve as a powerful platform for comprehensive identification and quantification of thousands of proteins and post-translational modifications (PTM) (Figure 1A,B). For example, LC–MS/MS-based methods are able to map reversible phosphorylation of serine, threonine, and tyrosine residues that serve as essential toggle switches regulating signal transduction pathways, such as those driving fibrotic remodeling [6]. Indeed, this technology is now used to routinely monitor thousands of molecules per experiment while maintaining highly specific and sensitive molecular measurements, owing to advancements in quantitative mass analysis and highly reliable, peptide-matching algorithms [14]. Modern MS instruments (e.g., Orbitraps) have sensitivity levels approaching attomoles, which allow them to distinguish proteins present in part per million within complex biomolecular mixtures such as total cellular lysates from cultured CMs [15,16].

The relative expression (recorded intensity values) of biomolecules detected by LC–MS/MS is usually normalized and organized into data sets, which are then analyzed with statistical tools to identify the most significantly changed proteins/phosphorylation sites between conditions, such as HCM samples versus healthy tissue. Univariate statistical hypothesis testing (e.g., Student’s *t*-test of means and ANOVA), coupled with a precise measure of abundance (e.g., log2 fold change), are simple methods for ranking and visualizing differential expression (Figure 1C). From these data processing methods, inferences at the single protein/phosphoprotein are made, such as upregulation of a particular growth factor or increased phosphorylation of an activation site on an important kinase or downstream substrate. Multivariate methods can be used to define a subset of features that best differentiate one group from another, revealing expression patterns (i.e., markers) within disease groups. For example, one can identify biomarker signatures by taking the most differential features (by statistical significance, biological plausibility, or both) via supervised or unsupervised clustering methods, and test if their relative expression can separate healthy and disease samples. For HCM, this was demonstrated in a recent proteomics study with plasma samples [17]. Sparse partial least squares discriminant analysis, a multivariate tool for collapsing high dimensional data into a lower-dimensional space, was used to find a subset of 50 proteins that best-distinguished HCM from case–controls. A smaller subset of these proteins correlated with known plasma indicators of advanced heart disease (e.g., troponin I), thus demonstrating the value of this exploratory analysis for expanding prognostic markers. In another analogous study, Sonnenschein et al. identified a set of four proteins downregulated in HCM based on known pathobiological function [18]. One of these downregulated proteins in HCM samples, c-kit (a tyrosine kinase), was singled out based on a hypothesized role in cardiac fibrosis and hypertrophy. When the corresponding gene was silenced in cardiac fibroblasts, researchers found marked upregulation of genes related to TGF-β signaling, a known mediator of cardiac fibrosis.

While automation of sample preparation and stable isotope labeling is ameliorating throughput, the shortcomings of proteomics are centered around relatively low sample capacity, high instrumentation cost and complexity, and lack of robust methods for single-cell assessment. Moreover, current “bulk” proteomic methods take an average measurement across all sampled cell populations, which muddles signals from distinct cell types. In the case of whole heart preparations, the proportion of CMs to specialized conduction cells can greatly influence the average protein signal measured. Emerging single-cell proteomic methods aim to add granularity to analysis and quantify cell population heterogeneity. Currently, single-cell analysis has been achieved by optimizing the efficiency of sample preparation, improving instrumentation sensitivity, and utilizing an isotopically distinct carrier protein “channel” to increase the detection of low-abundant proteins [19]. These aim to maximize the protein coverage from each cell in both DDA and DIA and maximize the number of protein matches between single-cell data to avoid zero values [20]. The latter aids differential analysis, while improved coverage provides better results for enrichment analysis. Strikingly, Brunner et al. were able to identify more than 800 proteins, from as low as 0.8 ng of cell extract, and indeed were able to quantify hundreds of differential proteins from even single cells transitioning through the cell cycle [21].

### 2.3. Pathway-Level Data Analysis and Functional Inference

While biosignatures have their particular role in research, a shared challenge for LC–MS/MS–omics is ascertaining the functional role of individual molecules in the complex signaling pathways of cells or their causal significance to disease progression. To address this issue, powerful systems using biology-based methods have been developed to identify higher-level biological pathways and cellular processes that are altered in a disease state from lists of differential biomolecules (Figure 1C). The most common statistical method for this is overrepresentation (enrichment) analysis, which tests if lists of differential features are overrepresented for a curated gene set. These sets are constructed from published data and comprise lists of proteins and genes (e.g., functionally coupled protein kinases and their downstream substrates implicated in a signal transduction cascade) [22,23]. There are annotated gene sets for nearly all well-known biological pathways and their corresponding phenotypic impact and can include proteins associated with diseases, interacting proteins, and putative kinase phosphorylation motifs. For instance, when Sonnenschein et al. measured differential protein expression following targeted inhibition of c-kit in cardiac fibroblasts, they performed gene set enrichment analysis to document downregulation of a novel extracellular matrix (ECM)–receptor interaction axis [18]. When combined with enrichment analysis, quantitative proteomics is exceptional for producing functional insights from global protein profiles, particularly when coupled with graphical software for organizing and navigating biomolecular networks, such as the Cytoscape data visualization environment [24] (Box 1).

Graphical analysis based on protein–protein interaction (PPI) networks offer an alternate framework for interpreting differential HCM expression data. As with other diseases, an initial genetic mutation may lead to a multitude of compensatory mechanisms and protein-level interactions that impact cellular phenotypes [25,26]. In this context, PPI networks can be instrumental for connecting a seemingly disparate array of disease-specific cardiac proteins to their respective topological relationships (e.g., components within a sarcomeric scaffold), ultimately elucidating their functional dependencies and explaining their impact on the pathobiological process. Direct PPI mapping studies can be performed in either a targeted or global manner, providing both qualitative and quantitative information (e.g., comparing association subnetworks altered between conditions).

One of the greatest challenges following this type of analysis is inferring pathophysiological relevance. While gene set annotations are useful for identifying differentially active biological pathways, enrichment of a pathway is not sufficient to establish mechanistic causality. For example, enriching the glycolysis pathway in HCM samples provides one line of evidence of increased glycolysis, but further validation is required to both measure activity of the pathway and determine its role in pathology. Moreover, these annotations may imperfectly cover components of these pathways, and one must carefully assess the differential features that cause the pathway to be enriched. Thus, candidate prioritization for validation must be carefully balanced between proposed biological relevance, statistical significance, and availability of selective validation tools (e.g., antibodies and targeted kinase inhibitors).

Given the depth of protein identifications achieved by modern LC–MS/MS systems and the public availability of well-established tools for bioinformatics analysis, it is no surprise that proteomics has emerged as an integral tool for understanding functional changes driving HCM. For example, in a recent proteomic profiling study of fibrotic remodeling in HCM, Kuzmanov et al. were able to quantify hundreds of differentially expressed phosphoproteins/kinases between affected and control samples, whose dysregulation was functionally validated to drive fibrosis [5]. Hierarchical clustering, principal component analysis (multivariate statistical analysis), and enrichment for curated kinase–substrate consensus sequences identified universal dysregulation of signaling by the protein kinase GSK3. This pattern was seen across organ-on-a-chip, septal myectomy, and murine samples. To validate the function of this protein kinase in fibrotic remodeling, targeted inhibitors of GSK3 were applied to organ-on-a-chip samples, resulting in a significant reduction in collagen deposition and myofibroblast activity. Overall, the application of quantitative proteomic profiling as a cross-platform approach proved a powerful strategy for generating functional insights, drug target selection, and compound screening.

Box 1Brief overview on reducing Type 1 error in proteomics.“Minimizing type I error in proteomics”Sample numbers are often low in proteomic studies, due to limitations in multiplexing and cost of scaling up throughput. Thus, it is essential to apply stringent multiple testing corrections to statistical tests in order to mitigate the issue of multiple comparisons. For example, let’s say we quantified 1000 proteins between treatment and control samples, the probability of at least one gene set or pathway being deemed significant at a defined *p <* 0.05 threshold due to chance (noisy data) alone is:


ℙ (making an error)=0.05



ℙ (not making an error)=1−ℙ (making an error)=1−0.05=0.95



ℙ (making at least one error)=1−ℙ (not making an error)



ℙ (making at least one error with a thousand tests )=1−ℙ (1−0.05)1000≅1


In other words, since there are many comparisons made in a proteomics experiment (both at the pathway and molecule level), it’s necessary to apply methods that decrease the Type 1 (false positive) error rate, which increases when many tests (e.g. student t-tests or gene set comparisons) are completed simultaneously. The most commonly used methods include Benjamini-Hochberg and Bonferroni corrections. Suitability and limitations of multiple comparison correction tests are extensively reviewed in Lualdi et al. [24].

## 3. Murine and Human Ex Vivo and In Vitro Models of HCM

As recently documented in Kuzmanov et al., omics analysis of HCM models have their own strengthens and weaknesses in terms of pathobiological relevance and ease of experimental implementation [5]. These systems can provide faithful representations of CM morphology and dysfunction in response to mutations in sarcomeric proteins and other molecular perturbations associated with HCM. Given the increasing adoption and relevance of cardiac models, it is evident that novel disease pathways and therapeutic leads can be discovered when in vitro and ex vivo systems are coupled with the omics profiling technologies highlighted in Section 2 of this review. While this section delves deeper into the recent developments in cardiac modeling, it is not exhaustive. For example, purified and synthetic filaments are indispensable for the biophysical characterization of subcellular changes in sarcomeres. However, since the focus of this paper is to review models that are most amenable to discovery science through high-throughput omics technologies, simplified biophysical systems are excluded from this article (Box 2).

### 3.1. Ex Vivo Models of HCM

#### 3.1.1. Murine Models

Murine models of HCM have existed for decades, utilizing a myriad of pioneering gene-editing technologies and surgical interventions to recapitulate the disease. One of the earliest models involved gene targeting, introducing a missense residue into the α-isoform cardiac myosin heavy chain (MHC, arginine substitution for glutamine, MHCR403Q) protein via a hit-and-run and recombineering method [27]. In mice, the α-isoform MHC protein (myosin-6, MYH6, gene) is homologous to human β-isoform MHC (MYH7 gene). While the predominant isoform expressed in fetal murine hearts is MYH7 and switches to MYH6 in adults, human hearts show the opposite trend [28,29]. Nevertheless, these mutant mice display many of the hallmarks of HCM, which, as in human patients, developed gradually. Young MHCR403Q mutant mice were comparable to wild type, but by 30 weeks, they showed atrial remodeling, myocyte disarray with enlarged hyperchromatic nuclei, and moderate fibrotic development. Moreover, the mutant hearts show altered physiological function, exhibiting discontinuous ventricular relaxation and decreased cardiac output, consistent with the diastolic dysfunction seen in HCM patients [2]. Early molecular studies leveraging the MHCR403Q mice to study the development of hypertrophy found an aberrant calcium-dependent mechanism, similar to the one driving pressure-overload induced hypertrophy [30]. These findings support current pharmacological paradigms, particularly the use of Ca^2+^ antagonists to mitigate diastolic dysfunction and arrhythmogenesis [31,32].

Other relevant mutations have been introduced into mouse strains by more traditional approaches, such as knock-out mutations of MYBPC3 and cardiac troponin [33,34]. Ferrantini et al. utilized such a transgenic model to make functional insights into contractile and calcium handling mechanisms in troponin mutants [33]. Moreover, rodents are amenable to viral transduction for delivery of CRISPR/Cas9 (and related technology) mediated gene editing for more targeted point mutations and the development of new genetic models [35]. Given the flexibility in creating murine models, these studies have been instrumental in mapping malignant mechanisms of HCM, especially when coupled with omics profiling approaches. In a recent proteomics study, Hu et al. analyzed the myocardia of obscurin-mutant mice to characterize the proteins driving an observed arrhythmic phenotype [36]. Their analysis revealed a mutation-specific upregulation of protein kinase cGMP-dependent type 1, a protein that modulates cardiac contraction via phosphorylation of cardiac ion channels. Furthermore, they found residue-specific increased phosphorylation of sarcoplasmic reticulum-associated proteins that facilitate excitation–contraction coupling.

Such murine models provide many advantages that can be exploited for omics discovery platforms. Principally, rodents have short gestation periods (approximately 20–30 days), well-established colony management methods, and defined genetic backgrounds (unlike an outbred patient population) [37]. Due to this fact, essential phenotypes such as hypertrophy can be exacerbated with chemical and physical treatments at the convenience of the researcher [38]. For example, transverse aortic constriction (a common surgical intervention for inducing hypertrophy) was adopted to expedite the development of hypertrophy in HCM mice [39]. Moreover, ease of handling allows researchers to harvest samples at various pathobiological and development stages. This is in contrast with patient samples. Despite human CMs having the highest clinical value, it is difficult to differentiate signal that is causative of the molecular HCM phenotype versus compensatory, since the natural history of HCM cannot be followed [38,40]. Patient biopsies are typically provided in the advanced stages of the disease, which is significant considering that many important clinical manifestations are not universally present in healthy HCM patients [1].

Another key benefit of murine models over other systems is that they provide mature primary CMs. Mature CMs are biologically complex, dynamic cells that execute a number of tasks beyond their primary contractile function, including peptide synthesis and extra-cellular communication [41]. As early as the perinatal stage, CMs are terminally differentiated and have undergone a number of ultra-structure morphologic and functional developments that make them distinct from the fetal versions [42]. For example, in a study measuring fetal murine heart maturation, Liu et al. combined RNA and protein profiling with electrophysiological recordings to demonstrate a shift in potassium channel subtype expression and density [43]. This developmental change in electrophysiology, a crucial component for stabilizing the resting membrane potential during the final stage of cardiac action potentials, resulted in increased activation kinetics and inward rectifying potassium current with respect to early development stages. Other parameters observed include changes in myofibril assembly (such as isoform switching in sarcomeric proteins), Ca^2+^ handling (maturation of the sarcoplasmic reticulum and expression of calcium channels), and metabolism [42]. Such differences result in separate phenotypes, which is important to consider when selecting an appropriate model, such as in the case of the more fetal-like hiPSC-CMs.

Murine models provide harvestable primary CMs, which can be directly measured by omics platforms (bulk tissue or sorted cells) or placed in tissue culture for further experimentation. An adult mouse heart can provide several milligrams of material, which is sufficient for LC–MS/MS-based identification and quantification of thousands of phosphoproteins and for performing rigorous differential analysis to investigate signaling cascades and other molecular alterations associated with cardiomyopathy [44]. Importantly, unlike hiPSC-CMs, these models provide whole heart preparations which capture data of the hemodynamic and multi-cell communication occurring in cardiac tissue that is missed by studying CMs in isolation. This is particularly the case for cardiac fibrosis, an insidious process observed in HCM patients. Non-CM cells play important roles in this process, from interactions with the ECM to paracrine signaling, which must be accounted for to holistically assess pathogenesis. Such mechanisms can be investigated in the ex vivo context via murine cm microtissue platforms composed of harvested cardiac cells. Alternatively, this process can be assessed via bulk tissue LC–MS/MS quantification, though individual cellular information cannot be distinguished from each other. Recent developments in single-cell proteomics technology and metabolomic cell profiles could further capture the crosstalk and response of individual cell types to their multicellular environment to create a more comprehensive profile of altered tissue systems in HCM.

Differences between murine and human hearts are well documented and include calcium handling, contractile protein isoform expression, and action potential kinetics [45]. Vakrou et al. further investigated this heterogeneity by quantifying HCM specific differences between the species [46]. This comprehensive molecular profiling included measurements of macromolecules and mitochondrial metabolism and found poor agreement between the tissue types. Differences in gene expression resulted in only two genes with similar expression values and no overlap in predicted signal pathways and transcription factors. These molecular differences were hypothesized to have profound effects on the cardiac phenotype. Thus, when utilizing murine models for discovery, additional experiments with human samples are recommended for follow-up validation.

#### 3.1.2. Primary Human Tissue

Explant human CMs are inarguably the gold standard for studying cardiomyopathy, given their biological relevance and physiological maturity. As discussed with murine models, there are numerous differences between mature and fetal CMs [47]. These include electrophysiological, morphological, and molecular expression differences, which result in distinct cellular phenotypes [48]. These differences are even more pronounced between species. Therefore, human CMs provide the nearest assessment of HCM while maintaining the advantages of an ex vivo system.

Innovative cardiac tissue dissection, CM isolation methods, and defined culture conditions have led to improvements in the quality and reliability of proteomic data generation. Although the availability of healthy human hearts is scarce, primary cells are usually procured from cryopreserved or fresh tissue donations, which allows for direct omics analysis or ex vitro culturing. Omics tools are apt for direct analysis of precious myomectomies since they require very small amounts of tissue. Advancements in tissue-harvesting protocols take advantage of this fact [49]. Grankvist et al. leveraged this by generating a novel instrument harvesting less than 1 mm of tissue [49]. Here, they found comparable RNA expression profiles between samples isolated using their submillimeter endovascular biopsy device and a conventional, larger-sized device. Moreover, they found decreased trauma to tissue, mitigating experimental artifacts while minimizing risk to patients as an added benefit. After collection, cardiac cells are either directly processed for the LC–MS/MS analysis or further purified to isolate and characterize CMs and other cardiac cells. The latter procedure has clear advantages since bulk tissue analyses contain mixtures of non-CM cells. Wojtkiewicz et al. described a primary CM isolation method for proteomic analysis, and quantitatively assessed CM purity before LC–MS/MS [50]. To ensure homogeneity, they coupled magnetic bead-based negative selection against contaminating immune and endothelial cells (CD45 and CD31, respectively) and flow cytometry to isolate CMs from stromal cells. This resulted in enrichment for greater than 80% titin-positive cells. Moreover, the isolated CM samples yielded 800 micrograms of protein starting from 250–500 milligrams of tissue, which is a respectable amount for phosphoproteomics, as well as 203 unique cardiac proteins not seen in the bulk analysis.

Despite limitations, functional information can be leveraged from bulk tissue measurements. For example, when analyzing clinical HCM tissue samples, Coats et al. identified downregulation of key cardiac metabolism pathways, particularly those involved with mitochondrial energy production [51]. These perturbations included downregulation of oxidative phosphorylation and lipid catabolic proteins, supporting the hypothesis of HCM pathology stemming from impaired energy production. Furthermore, when observing differential proteins, they identified an ECM proteoglycan, lumican, as significantly upregulated in HCM. Lumican modulates collagen assembly and is implicated in tissue fibrosis via TGF-β interaction [52]. In a similar experiment, Pei et al. applied proteomics, chromatin sequencing, and RNA sequencing to HCM and control tissue [53]. As with Coats et al., they found decreased enrichment of fatty acid metabolism via their proteomic data, which was corroborated by decreased RNA expression and hypoacetylation of key genes. Additionally, they observed an increased enrichment of an ECM modulation pathway. This agreement between these studies suggests that even without analyzing a purified CM population, there are consistent findings between studies.

CMs are available for culturing prior to the LC–MS/MS analysis, which is useful for functional and overexpression studies. CMs can be purchased commercially, and there are publicly available protocols for isolating CMs from tissue [54]. Although terminally differentiated, human CM cells can be passaged and are viable in culture for days. Nevertheless, CMs experience morphological and electrophysiological changes after prolonged culture which can impact functional studies. For instance, Li et al. observed a near loss of sarcomeres in adult CMs by day 10 in culture [55]. Timing is thus imperative for minimizing adaptions to culture.

Regardless of the methods used prior to the LC–MS/MS analysis, this model has the advantage of capturing the dynamic signaling events of the cardiac microenvironment. As with murine models, these interactions can be captured from whole heart extracts as well as isolated cells, allowing one to detect modulations to CMs that are missed by monoculture. Pathway and PPI network analysis offer a causal framework for mapping changes in these signaling axes, particularly the discovery of signaling proteins such as cytokines destined for secretion. Maron et al. demonstrated the power of these computational tools by generating individualized PPI networks from HCM patient myectomy samples [56]. Their analysis identified that a significant portion of HCM patients enriched for fibrosis pathways, compared with controls, which was validated by increased interstitial collagen in these samples. Using this pathway information, they identified a patient-specific subset of interacting genes (from the top differential HCM features) that predicted extreme fibrotic phenotypes in a group of patients. Overall, such tools can be used to identify networks of interactors which can help inform clinical decisions such as patient outcomes and eventually lead to functional insight.

Box 2Current advancements in integrated proteomics and metabolomics.“Emerging Integrative Omics: Spotlight on metabolomics”Metabolomics aims to quantify low molecular weight molecule substrates and products involved in metabolic processes . These metabolites are the end-products and mediators of protein expression, such as inositol trisphosphate and lactate, which are essential regulators of signal transduction and metabolism in CMs. As with proteomics, LC/MS is a technology platform of choice to measure metabolites on a large-scale.Proteomics and metabolomics can act synergistically for to making make functional insights through joint pathway analysis [52]. First, there are robust experimental methods for extracting metabolites and protein/phosphoproteins from specimens simultaneously with minimal losses. For example, liquid-liquid extraction (LLE) procedures can be used to selectively separate metabolites and proteins by differences in solubility, with subsequent solid-phase microextraction (SPME) allowing for purification of metabolites with minimal matrix interferences that confound LC/MS [53].Once metabolites are identified and quantified, specific metabolic enrichment analyses allow researchers to interrogate changes in metabolic pathway activity via annotated enzyme-metabolite interaction networks [54]. Recent advancements in database searching also allows metabolism-specific searches with proteomics data [55,56,57,58,59,60]. This way, complementary iterations of metabolism-related enrichment analysis (one for metabolites, one for enzymes/proteins, and one combining both data) that directly link alterations in enzyme levels or PTMs in reaction-driven biochemical pathways to changes in metabolite levels.

### 3.2. In Vitro Models of HCM

Primary cardiac cells are minimally proliferative once harvested. In contrast, hiPSCs offer near-unlimited and relatively consistent sources of human CMs for disease modeling and have been used extensively for proteomic research [61]. These hiPSC-derived CMs are differentiated by manipulating key signaling pathways implicated in cardiac development, resulting in immature but functional fetal CMs with hallmarks of adult CMs [62]. In the case of HCM, hiPSCs harboring site-specific alleles of nearly all major disease-associated mutations have been created, primarily generated from HCM patient-derived stromal cells (typically reprogrammed dermal fibroblasts) or, more recently, by introducing locus-specific HCM mutations into hiPSC lines using CRISPR/Cas9 editing technology [63,64]. Both routes provide a unique opportunity for global omics profiling of maladaptive mutation-associated pathway alterations, and subsequent drug screening to modulate patient/mutation-specific phenotypes.

Before reviewing published research, it is essential to address the clear limitations of these models. In many respects, such cells are equivalent to fetal stage CMs and lack many of the (electro)physiological and cellular functions of their mature, adult counterpart [62]. For instance, the relevant ultra-structures for regulating Ca^2+^ (e.g., the T tubules and the sarcoplasmic reticulum), are absent or underdeveloped in hiPSC-CMs [62]. This is reflected by lower transcription of genes encoding the functional protein components belonging to these structures [65,66]. Other deficiencies, such as fetal-like metabolism, have been described extensively [62,67]. Despite limitations, these cells are inarguably valuable for cardiac modeling given their accessibility and relevance to human biology. Moreover, incremental advancements toward cardiac cell maturation have been described, such as increased culture time, in vivo maturation, and biophysical/hormonal stimulation [62,68,69]. Chemical stimulation can enhance the development of functional T-tubule networks, alleviating issues of excitation–contraction coupling [70]. Furthermore, innovative 3D tissue reconstitution and in vivo maturation (hiPSC transplantation) hold great promise toward elevating maturation deficiencies [62,71].

Mutant hiPSC cm exhibit many of the cellular hallmarks of HCM, particularly when engineered into a cardiac microtissue context, making them excellent models for investigating pathophysiology via exploratory omics analysis. Myofibrillary disarray and hypertrophy, two key phenotypes, were reported in a number of studies modeling different contractile protein mutations [63,72,73,74]. When evaluating a novel nanopatterning approach for producing adult-like CMs, Pioner et al. measured both increased cell area/perimeter and myofiber disarray in engineered CMs expressing myosin heavy chain mutations [75]. Similarly, Tanaka et al. saw a modest but significant increase in surface area and ultrastructure disarray [74]. Other important physiological changes, such as aberrant Ca^2+^ handling (hypothesized to play a role in diastolic dysfunction), increased mitochondrial metabolic activity, and mutation-specific hypo- and hypercontractile force have been documented [63,73,76,77].

Despite the successful application of proteomics with a number of different cardiomyopathy models, there are surprisingly few studies utilizing hiPSC-CMs for in-depth functional proteomic assessment of HCM. Omics technology can be applied to such models to investigate the exact molecular perturbations that drive observed functional changes. For example, proteomics can reveal the signal transduction cascades and regulatory proteins that cause abnormal intracellular Ca^2+^ concentrations, such as the specific phosphosites on effector protein targets downstream of calcium–calmodulin-dependent protein kinase II [78,79]. Equally, a combined proteomics and metabolomics analysis could be used to elucidate the mechanisms driving increased mitochondrial respiration, such as metabolites and protein pathways involved in ATP production [63]. For example, Hellen et al. found temporal metabolic protein expression patterns when measuring hiPSC-CM development via proteomics [80]. These included an increase in oxidative phosphorylation-related proteins over culturing time.

hiPSC-CMs avoid many of the disadvantages of in vitro culturing of primary CMs, which makes them excellent for high-throughput omics analysis. hiPSCs are viable in cell culture for long periods as undifferentiated cells and can be differentiated into CMs as needed [81]. As discussed in the primary CM section, adult CMs have limited proliferation and exhibit a number of changes with prolonged culture. These changes are noted at the superstructure level which will unquestionably affect the proteome and metabolome measured. Due to their defined culturing advantages, hiPSC-CMs have greater flexibility for experimentation, such as in the case of drug treatment regimens [82]. Moreover, because the hiPSCs can be expanded prior to differentiation, these cells can yield much higher amounts of cellular material with respect to primary CMs. This is essential for phosphoproteomic analysis since phosphorylation sites are often sub-stoichiometric, hence significantly less abundant than unmodified proteins, and require enrichment for detection. The enrichment techniques also have varying efficiencies and thus require different starting peptide concentrations, with the most efficient methods (e.g., immobilized metal affinity chromatography) requiring several hundred micrograms to milligrams of protein extract per sample [83,84]. Another strength of hiPSC-CMs for HCM studies is that they provide plentiful material as isogenic control, which is essential for measuring cellular responses driven by mutation rather than spurious differences in genetic background. As mentioned above, hiPSC-CMs can be used to study practically all known HCM-associated mutations, allowing for personalized pathobiological assessment.

### 3.3. Future Advancements in Tissue Modeling

Researchers are increasingly pivoting to new 3D systems to better recapitulate important cell–cell communication, which are likely the major drivers of pathobiology in cardiomyopathy. These systems, referred to as engineered heart tissue (using scaffold matrix) or cardiac spheroids (spontaneous assembly), are constructed from cardiac cells such as CMs, fibroblasts, and endothelial cells [85]. When assembled in appropriate cellular proportions, they develop similar functional characteristics to cardiac tissue. These include vascular network formation, spontaneous CM contraction, and ECM assembly [86]. These platforms have been constructed with hiPSC-CMs and primary murine and human CMs, with hiPSC-CMs having a distinct advantage since cell death is a major component in platform fabrication efficiency [86,87]. Interestingly, hiPSC-CMs in 3D co-cultures exhibit greater maturation with respect to 2D monoculture [71,88]. Polonchuk et al. found structural and functional similarities between cardiac spheroids of hiPSC and primary CM origin [86].

The omics data captured from 3D co-culture systems represent the nearest version of molecular profiling of tissue in a dish. Molecular data from these systems fully integrate multicellular interactions within the cardiac microenvironment, such as integrin-based mechanotransduction and paracrine signaling. Here, the HCM-specific mechanisms of hypertrophy, fibrosis, and microvascular occlusion can be directly quantified for hypothesis generation, and functionally validated within the same platform (Figure 1C). Moreover, high-throughput drug interactions can be assessed similar to as reported by Polonchuk et al. [86]. In their study, they used 3D tissue models for drug toxicity modeling and screening and discovered a novel, fibroblast-induced toxicity from doxorubicin (a known cardiotoxic drug). Alternatively, these platforms can be adopted to comparatively study the consequences of specific sarcomeric protein point mutations and to elucidate the molecular basis of variable phenotypes in patients (e.g., asymptomatic hypertrophy vs. fibrotic remodeling). Undoubtedly, these technologies have transformed our approaches toward studying cardiomyopathy and will serve as important tools for delivering precise, patient-specific care in HCM.

## 4. Conclusions

HCM research is poised for comprehensive molecular characterization and high-throughput drug discovery through the breadth of powerful new quantitative omics platforms and cardiac models presented in this review. The emergence of allied computational tools for hypothesis generation, such as pathway analysis and PPI networks, allows one to distinguish causal mechanisms driving the diversity of clinical phenotypes observed. Complementing these tools is the breadth of HCM models discussed in this review. Murine samples are an economic model providing accessible primary mature CMs, in contrast to clinical specimens, which are hard to access and typically represent end-stage disease. Yet, there are clear species-specific differences that limit the generalizability of findings. Currently, hiPSC-CMs are the intermediate between the clinical relevance of patient samples and the accessibility of murine samples. Ultimately, optimization of novel 3D systems will serve as a bridge between models. These systems promise to advance bench-to-bedside research by recapitulating important pathobiological features of HCM.

Nevertheless, there still remains a gap between discovery and functional research. Careful collaboration between bioinformaticians, molecular biologists, and clinicians will be necessary to address this challenge and create new platforms for deriving meaningful information from LC–MS/MS data.

## Figures and Tables

**Figure 1 ijms-22-13644-f001:**
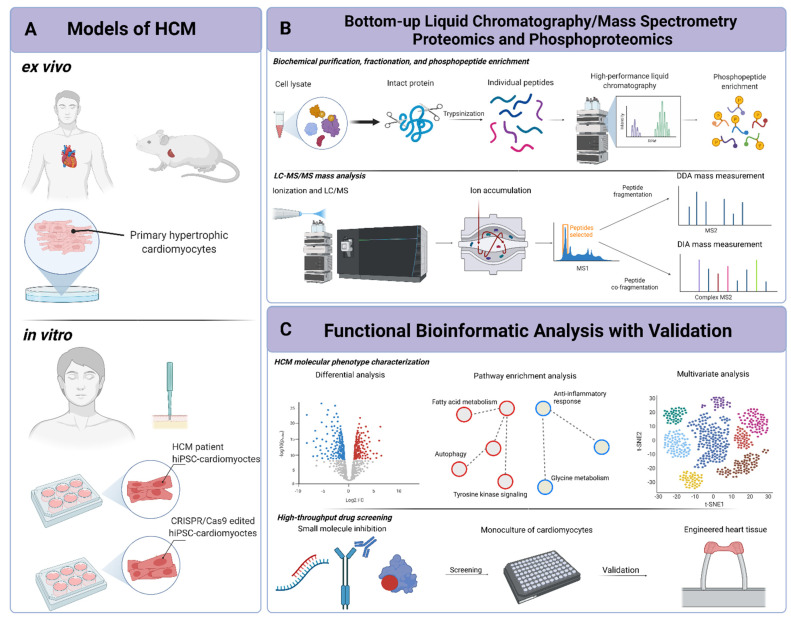
In vitro and ex vivo models of hypertrophic cardiomyopathy (HCM) and phosphoproteomic workflow: (**A**) cardiac cells are obtained from ex vivo (human or murine) or in vitro (human-induced pluripotent stem cells, (hiPSCs)) sources prior to phosphoproteomic analysis. In vitro cardiomyocytes, differentiated from a patient harboring an associated HCM mutation, or cells that have been CRISPR/Cas9 edited prior to differentiation; (**B**) once harvested, proteins are isolated from total cell lysate (homogenate) via a bottom-up phosphoproteomic method. Extracted proteins are proteolytically cleaved with a site-specific protease (e.g., trypsin) into peptides, purified by hydrophobic (reversed-phase) resin, and subject to high-performance liquid chromatography for fractionation. Phosphopeptides are then enriched by immobilized metal affinity chromatography. The total peptides and enriched phosphopeptides are then analyzed, separately, by liquid chromatography–mass spectrometry (LC–MS/MS), either by data-dependent or data-independent analysis (DDA and DIA, respectively). In DDA, high abundant peptides are selected individually during ion accumulation and fragmented separately, while in DIA, multiple co-eluting peptides are co-fragmented in a time window resulting in more complex (mixed) fragmentation spectra; (**C**) the resulting spectra are analyzed computationally to identify the corresponding proteins and to make functional insights into up- and downregulated proteins, phosphoproteins, and the likely activity changes in their corresponding biochemical pathways. Analyses include univariate and multivariate statistical tests. Finally, exploratory findings from phosphoproteomics are validated, typically by small-molecule inhibition, both with high-throughput monoculture and novel engineered heart tissue platforms.

## Data Availability

Not applicable.

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
