# Peer review of "Mass-Spectrometry-Based Functional Proteomic and Phosphoproteomic Technologies and Their Application for Analyzing Ex Vivo and In Vitro Models of Hypertrophic Cardiomyopathy"

_ijms, 2021, doi:10.3390/ijms222413644_

Round 1
Reviewer 1 Report
I think this paper is important.
However, I think it is better to use expressions that are easier for the reader to understand.
The sentences are long and difficult to understand Please summarize the points using a table.
Are there any other limits to this research?
Textboxes 1 and 2 should be summarized as a table or figure.
Reviewer 2 Report
The idea of this study, to evaluate state-of-the-art mass spectrometry-based proteomics and phosphoproteomics technologies and their application to in vitro and ex vivo models of HCM, is very interesting and with important future implications. This is generally a well-written review, with a clear central figure and interesting findings that can make significant contributions to clinical practice. However, I suggest to better highlight the limitations of the study. Also, given that there are other recent studies that are very similar to this one, I recommend highlighting the original elements of the study. Also, I suggest better summarizing the study’s conclusions because in this form they do not address the main question posed. Also, the article is not very easy to read and I consider that the manuscript would benefit enormously from English editing.
